# Population structure reverses selection of variants with proportionally scaled birth and death rates

Natalia L. Komarova [1] ✉ & Dominik Wodarz [2] ✉

A widespread biological phenomenon is that higher reproduction rates are often accompanied by higher mortality. During tumor progression, variants can both reproduce and die faster; rapidly replicating viruses decay more quickly; arthropods with faster reproduction have shorter lifespans; and in ecological systems, more frequent reproduction can increase predation risk. Variants with proportionally scaled birth and death rates are termed *quasi-neutral mutants*. Although their lifetime reproductive success is unchanged, such mutants have fixation probabilities slightly lower (or higher) than neutral mutants if birth and death rates are proportionally larger (or smaller). Previous studies, limited to well-mixed populations, showed that quasi-neutral mutants deviate from neutrality but still exhibit fixation probabilities scaling with their initial frequencies. Here, we show that in deme- or spatially structured populations, variants with proportionally increased (decreased) birth and death rates become genuinely disadvantageous (advantageous). We calculate their effective fitness and further demonstrate that even when mutants have higher lifetime reproductive output, proportional increases in both birth and death rates can render them strongly disadvantageous—and vice versa. This effect intensifies in larger populations. These findings revise the relationship between lifetime reproductive success and selection, with implications for evolutionary dynamics across biological systems.

The emergence of advantageous mutants is often driven by selection for increased reproductive rates, allowing certain variants to outcompete others within a population. Faster reproduction, however, is rarely without trade-offs. Increased proliferation often comes at the cost of reduced survival, higher metabolic demands, or greater susceptibility to environmental stress[1,2]. This has been observed in microscopic systems, such as viruses, bacteria, or tumor cells, as well as in higher organisms, such as arthropods, fish, dolphins, snakes, birds and lizards among others[3–10]. If the overall outcome of these tradeoffs leads to a larger life-time reproductive success, such a variant is conventionally thought to have a selective advantage.

In the recent years, it has become apparent that the interplay between birth and death rates can influence the evolutionary dynamics in subtle ways, particularly when these kinetic rates are proportionally scaled across competing variants. A striking example of this phenomenon is the behavior of *quasi-neutral mutants*—variants whose birth and death rates are scaled by a common factor, preserving their maximum lifetime reproductive output but altering their turnover rates. Parsons et al.[11] studied a birth-death process with demographic fluctuations, and demonstrated that in well-mixed populations, quasi-neutral mutants exhibit fixation probabilities that deviate from strict neutrality. In particular, faster-turnover mutants are less likely to fixate due to increased demographic stochasticity, and slow turnover

[1]Department of Mathematics, University of California San Diego, La Jolla, CA, USA. [2]Department of Ecology, Behavior, and Evolution, University of California San Diego, La Jolla, CA, USA. ✉e-mail: nkomarova@ucsd.edu; dwodarz@ucsd.edu

mutants are more likely to fixate. This finding challenged classical neutral theory by showing that even in the absence of fitness differences, proportional scaling of kinetic rates alone could shape evolutionary outcomes.

Similar patterns were observed in other processes. For example, the role of cell cycle acceleration and deceleration in evolutionary dynamics was studied[12]. Two life strategies were compared: repair-profficient cells that enter temporary cell cycle arrest (corresponding to repair) and repair-deficient cells that do not arrest (and thus cycle faster). It was found that although non-arresting cells were characterized by a faster growth in isolation, this did not translate into a selective advantage in the model: faster cycling cells fixated with a probability that was lower than predicted for a neutral scenario. It was further shown that similar patterns are observed in a death-birth Moran process, where faster (slower) mutants fixated at a lower (higher) rate compared to neutral.

A recent paper by Bhat and Guttal[13] offers a comprehensive study of the interplay between Malthusian fitness and turnover in well-mixed populations. Their analysis revealed how noise-induced selection can reverse the direction of expected evolutionary trajectories. The authors derive stochastic differential equations from a general non-linear birth-death process to describe changes in population densities and trait frequencies, revealing a drift term containing a balance between natural selection for increased ecological growth rate and a stochastic selection for reduced variance in changes in population densities, which favors species with a lower turnover.

The latter mechanism can reverse the direction of selection predicted by deterministic models, particularly in small populations or under weak selection. This approach highlighted the dual mechanisms by which demographic stochasticity biases evolutionary outcomes: directly, by favoring the variants with a higher Malthusian fitness, and indirectly, through noise-related effects that depend on the turnover rate. The work further generalized classical equations like the replicator-mutator equation, Price equation, and Fisher's fundamental theorem to finite populations, providing a framework to understand how stochasticity interacts with natural selection and drift.

In this study, we build on these theoretical advances by exploring how population structure, such as deme or spatial organization, further modulates the evolutionary fate of mutants with proportionally scaled kinetic rates. While previous work focused on well-mixed populations, we demonstrate that structured environments fundamentally alter selection pressures. Variants with proportionally increased birth and death rates, which are quasi-neutral in well-mixed settings, become truly disadvantageous in structured populations, and vice versa for decreased rates. Moreover, we show that even mutants with a higher lifetime reproductive output can be rendered disadvantageous by accelerated turnover, an effect that in spatially structured populations grows with the population size.

## Results

Suppose that the life-time reproductive success of a variant is changed by a certain percentage. Can a concomitant increase of death and birth rates influence, or even reverse, the selection of this variant? To answer this question, we first consider the concept of "quasi-neutral" mutants[11,13,14]. Quasi-neutral mutants are defined by having proportionally scaled birth and death rates compared to the wild-type, i.e., $r_m = \tau r_w$ and $d_m = \tau d_w$, where $r$ denotes the reproduction rate, $d$ the death rate, subscripts $w$ and $m$ denote wild-type and mutant individuals, and $\tau$ is a constant acceleration or deceleration factor, which we will refer to as a turnover factor. Although the maximum total reproductive output during the life-span of the individuals is identical for wild-type and mutant populations, these mutants' probability of fixation is not simply given by their initial frequency. Assuming a well-mixed population of mean size $N$, the fixation probability of a quasi-neutral mutant for stochastically fluctuating (Verhulst-type)

populations under the diffusion approximation[11,13,14] is given by $p_{fix} = \frac{1}{N} \times \frac{2}{\tau+1}$ (remarkably, the same expression describes the exact probability of mutant fixation is a death-birth Moran process[12]). In other words, mutants with a faster turnover ($\tau > 1$) have a lower fixation probability compared to neutral mutants, and mutants with a slower turnover ($\tau < 1$) have a higher fixation probability. The reason is increased demographic fluctuations in the faster turning type compared to the slower type. These mutants, however, do not behave like truly disadvantageous (advantageous) variants because their fixation probability only differs from $1/N$ by a constant (in $N$) factor. Hence, the term quasi-neutral.

These evolutionary dynamics change drastically in structured populations. Consider a deme model with logistic birth-death dynamics in each deme[15], and where divisions are density-regulated and can also result in the offspring individuals migrating to other demes, regardless of their location (see Methods). Now, the value of $N_{tot} \times P(\text{fix})$ for faster-turnover mutants declines exponentially with $N_{tot}$ (Fig. 1A), which is the characteristic of truly disadvantageous mutants. In other words, although the life-time reproductive success of the higher turnover mutant individuals is identical to that of wild-types, they experience negative selection. The extent of the disadvantage the mutant experiences becomes larger with the amount by which the turnover is increased relative to the wild-type (parameter $\tau$, Fig. 1B), as well as for larger death-to-birth ratios of the individuals (Fig. 1C). For mutants characterized by a lower turnover compared to the wild type ($\tau < 1$), as the population size increases, the probability of fixation increases and saturates at a constant level, indicating that these types of mutants experience a true selective advantage, see Supplementary Fig. 4. Therefore, the deme population structure fundamentally changes the properties of mutants in which birth and death rates are proportionally scaled while keeping the reproductive output constant.

To explain this behavior, we used a coarse-grained approach[16], which is applicable when the mutation rate is sufficiently low such that individual demes are typically homogeneous with respect to the type (either fully wild-type and fully mutant, see the schematic in Fig. 1D). In this case, the process is reduced to a 1D Markov walk, governed by a deme conversion process, whereby an individual of one type migrates to a random deme of a different type and fixates successfully in the target deme (see Supplementary Note 1). Let us denote by $\bar{\rho}_m$ the expected probability of a single mutant to fixate in an isolated wild-type deme at quasi-equilibrium, and $\bar{\rho}_w$ the expected probability of a single wild-type individual to fixate in an isolated mutant deme. Then the probability of mutant fixation in a system of $D$ demes, starting from a single mutant individual, is given by $\Pi_1^{frag} \approx \bar{\rho}_m \frac{1-\frac{1}{\mathscr{F}}}{1-\frac{1}{\mathscr{F}^D}}$, where $\mathscr{F}$ is the relative fitness of a mutant deme, defined by the ratio of two deme conversion rates, from wild-type to mutant and from mutant to wild-type. For quasi-neutral mutants, $\mathscr{F} = \tau \frac{\bar{\rho}_m}{\bar{\rho}_w}$. Figure 1A–C shows excellent agreement between this theory and computer simulations.

Note that the size of the effect depends on the degree of fragmentation and gets stronger as the population is split into a large number of small demes, see Fig. 1E and S5. Suppose the total population of size $N_{tot}$ is split into $D$ demes of size $N$ each. Decreasing individual deme size $N$ (and therefore increasing $D$) will magnify the effect, thus making accelerated (decelerated) mutants less (more) advantageous. If, on the other hand, the demes are large, the effect disappears: for large $N$, the deviation from the diffusion approximation is negligible, and we have $\frac{\bar{\rho}_m}{\bar{\rho}_w} = \frac{1}{\tau}$, thus resulting in $\mathscr{F} = 1$ (deme neutrality) and $\Pi_1^{frag} = \frac{1}{DN} \times \frac{2}{\tau+1}$, which is exactly the same as the probability of fixation in a non-fragmented system (of size $N_{tot} = DN$). In other words, fragmentation into large demes does not lead to our reported results.

In the case of smaller demes of equilibrium size $N$, we can define an effective fitness parameter for quasi-neutral mutants, $f_e = \mathscr{F}^{1/N}$ (see Fig. 1F). Then, for the probability of mutant fixation in a fragmented

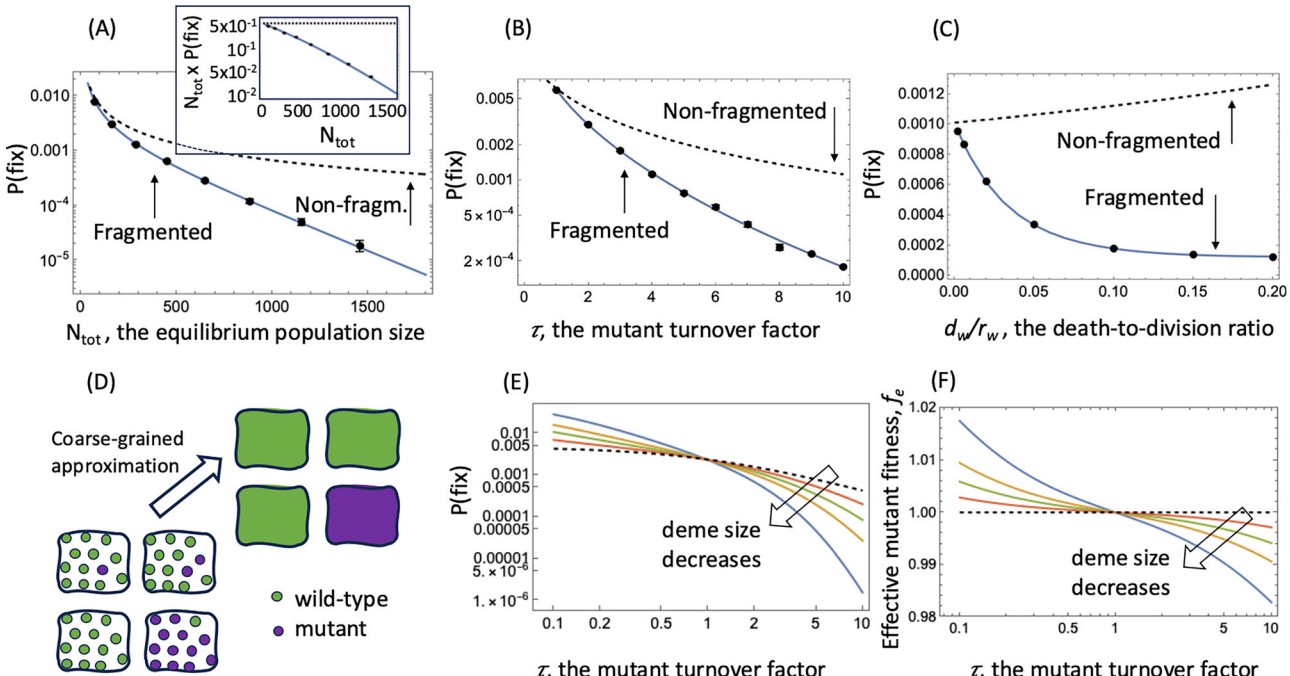

**Fig. 1 | Properties of a mutant with proportionally increased birth and death rates (expressed by turnover factor $\tau > 1$) in the deme-structured model.** A–C The points show results from computer simulations, with the bars representing the binomial proportion standard error (too small to see in some cases), based on 1,032,000 simulations per point. The solid line is the coarse-grained theory prediction. The dashed line indicates the theory predictions for quasi-neutral mutants in a non-fragmented population. **A** The mutant fixation probability, P(fix), declines exponentially with the total quasi-equilibrium population size, $N_{tot}$, a characteristic of a disadvantageous mutant. Inset: the quantity $N_{tot}$ x P(fix) as a function of $N_{tot}$ is a constant in non-fragmented populations, and declines in fragmented populations. **B** The mutant fixation probability declines substantially with the amount by which the birth and death rates are proportionally increased, $\tau$. **C** The reduction of the mutant fixation probability in the deme-structured compared to the non-spatial population becomes more substantial with higher death to birth ratios. Base parameters were chosen as follows: For (**A**) $\tau = 2$; D varies from 4 to 100. For (**B**) $D = 9$. For (**C**) $\tau = 10$, $D = 9$. **D** A schematic showing the concept of a coarse-grained approximation. **E** The probability of mutant fixation (obtained by the coarse-grained approximation) as a function of $\tau$, in a deme-structured model with the total carrying capacity 480, split into equal demes of size $K = 20,30,40,60$. **F** The effective mutant fitness, $f_e$, corresponding to the same deme systems as (**E**). The rest of the parameters, unless otherwise noted, are $d_w / r_w = 0.1$; $\epsilon = 10^{-4}$; $K = 20$; s = 0.

population, we have $\Pi_1^{frag} = \alpha \times \frac{1-1/f_e}{1-1/f_e^{N_{tot}}}$, where $\alpha$ is a constant independent on the deme number. Effective fitness quantifies selection pressures resulting from fragmentation: $f_e < 1$ for faster mutants, and $f_e > 1$ for slower mutants, which explains the numerical observations described above.

A biologically more realistic scenario is to consider a mutant that has a higher or lower life-time reproductive output compared to the wild type, but also has accelerated/decelerated birth and death rates (altered turnover). Assume that division and death rates of mutants are given by $r_m = \tau (1 + s) r_w$ and $d_m = \tau d_w$, where $s$ (with $s > -1$, and $|s| \ll 1$ for many applications) is the selection coefficient. If $s > 0$, the life-time reproductive success of the mutant is larger than that of the wild-type (traditionally considered an advantageous mutant), and for $s < 0$, the life-time reproductive success of the mutant is smaller than that of the wild-type (traditionally considered a disadvantageous mutant). Figure 2A plots the mutant fixation probability as a function of the fold increase in mutant turnover, $\tau$, for a scenario where $s = 0.01$. For $\tau = 1$ (no change in turnover), the fixation probability is well predicted by the established theory, where for large populations ($1/N \ll s \ll 1$) it is approximately given by $s$ (see Supplementary Note 1, Section 1.1.2 and ref. 17 for constant-population processes, and Parsons et al.[11] for the extension to populations with demographic stochasticity). As the mutant turnover, $\tau$, is increased relative to the wild-type, however, the fixation probability declines sharply and becomes lower than $1/N$, the fixation probability of a neutral mutant. In other words, by proportionally increasing the birth and death rate of a mutant (turnover parameter $\tau$), the mutant turns from having a selective advantage to

having a strong selective disadvantage. Conversely, a mutant that is disadvantageous for $\tau = 1$ (e.g., $s = -0.01$) can become advantageous if it has proportionally reduced birth and death rates ($\tau < 1$, Fig. 2B). In Supplementary Note 1 we provide general expressions for the deme fitness, $\mathscr{F}$, for any type of mutants, under a variety of model assumptions, including different models of migration and mutant initialization. Figure 2A,B again demonstrates excellent agreement between the coarse-grained theory and computer simulations in a deme model.

Remarkably, the effect of selection reversal described here increases with the total population size. In this context, it is interesting to compare it with selection reversal effects that have been previously reported. In a previous study[18] it was found that in a death-birth Moran process, mutants with proportionally scaled division and death rates fixated at a rate that is lower (higher) than neutral if they experience acceleration (deceleration) of the kinetic rates; this effect disappeared as the population size increased. In a recent paper[13], selection reversal was investigated comprehensively in (non-fragmented) systems with demographic stochasticity, and while the presence of turnover acceleration (deceleration) provided a term that countered the effect of Malthusian selection, its relative contribution scaled with the inverse system size.

Figure 2C, D demonstrates the qualitatively different behavior of the fragmented system (red) as the population increases, compared to the non-fragmented system of the same size (blue). The two coincide at the leftmost point, which corresponds to a single deme of $K = 20$, and diverge as the total population size increases (in the case of the fragmented population, the number of demes increases). In panel (C),

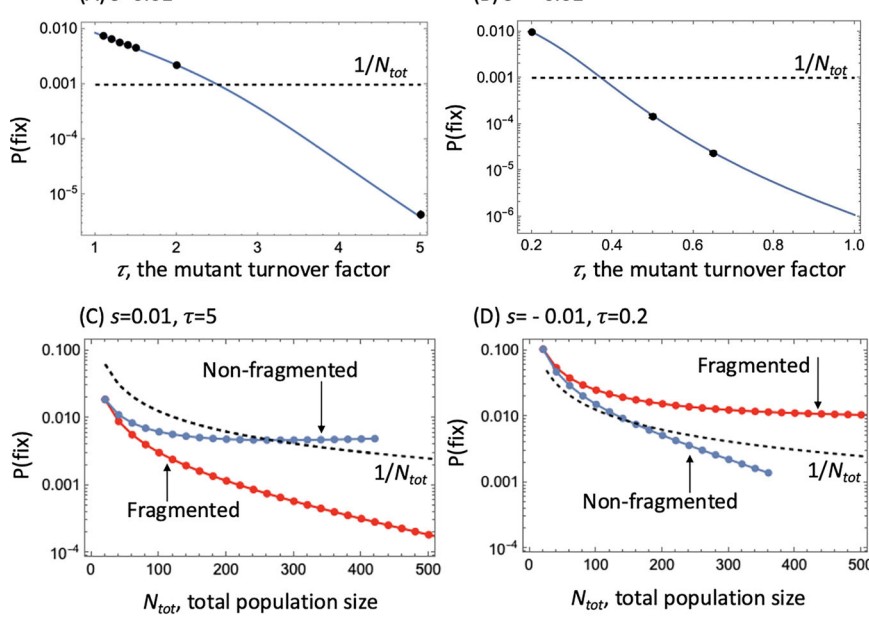

**Fig. 2 | Properties of a mutant with proportionally increased birth and death rates (expressed by turnover factor $\tau$), and additionally a changed life-time reproductive output compared to the wild type in the deme-structured model.** The changed life-time reproductive output of the mutant is expressed by the selection coefficient, $s$, such that $r_m = \tau (1 + s) r_w$. Mutant fixation probabilities, P(fix), are shown as function of $\tau$, the mutant turnover factor. **A**, **B** The points show the mean of computer simulations, with the bars representing the binomial proportion standard error based on 1,032,000 runs (too small to see). The solid line is the coarse-grained theory prediction. The dashed line represents the fixation probability of a neutral mutant, $1/N_{tot}$ where $N_{tot}$ is the total population size. **A** A mutant

that has a 1% higher life-time reproductive output compared to the wild-type is advantageous for $\tau=1$ and small values of $\tau$, but turns disadvantageous for higher mutant turnover rates (larger $\tau$). **B** A mutant that has a 1% lower life-time reproductive output compared to the wild-type is disadvantageous for $\tau=1$ and higher values of $\tau$, but turns advantageous for lower mutant turnover rates (smaller $\tau$). **C**, **D** Comparison of a fragmented (red) and a non-fragmented (blue) system of the same size (coarse-grained theory). The probability of fixation is plotted as a function of $N_{tot}$, for (C) $s=0.01$, $\tau=5$ and (D) $s=0.01$, $\tau=0.2$. For non-fragmented systems, reversal of selection disappears for larger system sizes; for fragmented systems it becomes stronger.

the selection coefficient is positive, but the turnover factor $\tau > 1$, resulting in a decrease of the probability of mutant fixation. Under the specific parameter values chosen here, for small population sizes, we observe reversal of selection for both fragmented and non-fragmented populations. That is, for the mutants, P(fix)<$1/N_{tot}$ despite their lifetime reproductive output being greater than that for the wild-types. While in non-fragmented populations the reversal disappears as $N_{tot}$ increases (with P(fix) crossing the $1/N_{tot}$ line and approaching a constant as $N_{tot}$ increases), the effect only becomes stronger in fragmented populations, showing an exponential decline of P(fix) with $N_{tot}$, which is a signature of disadvantageous mutants. Similarly, Fig. 2D explores the case with a negative selection coefficient and a turnover factor $\tau < 1$. For small population sizes, selection is reversed (P(fix)>$1/N_{tot}$), but this effect disappears for non-fragmented systems (with P(fix) decreasing exponentially below the $1/N_{tot}$ line). At the same time, for fragmented populations P(fix) remains greater than neutral and in fact starts approaching a constant level, consistent with the behavior of advantageous mutants.

The reversal of selection force is especially striking in settings where wild-type individuals generate mutants at a constant rate $\mu$. Figure 3A, B shows that a mutant that is advantageous ($s = 0.01$) for $\tau = 1$ invades the deme-structured population as expected, but an accelerated mutant with the same relative advantage ($\tau > 1$) persists at a selection-mutation balance, a characteristic of a disadvantageous mutant. Figure 3C, D shows a similar simulation for $s = -0.01$. Without increased turnover ($\tau = 1$), the mutant is disadvantageous and persists at a selection-mutation balance. For $\tau < 1$, however, the mutant takes over, which is characteristic of an advantageous mutant. The coarse-grained approach allows for an analytical estimate of the selection-mutation level and the threshold fold-difference in the turnover rate that reverses selection (by making advantageous

mutants disadvantageous and the other way around), see Supplementary Note 2.

Finally, the same trends are observed in a spatially explicit agent-based model, although the effect is weaker (Supplementary Note 3). The similarity of results in spatially explicit models and deme-structured models with non-spatial migration indicates that population fragmentation is a key driver of these dynamics.

Note that we have chosen to analyze the density-dependent division model because it closely matches the spatially explicit simulations; it also is arguably a more biologically realistic assumption as cells slow down their divisions in crowded conditions. One way to generalize the results of the fragmented model is to assume that the per capita division rate is $rb(x)$ with $b(0) = 1$, $\frac{db}{dt} \leq 0$, and the per capita death rate is $d\delta(x)$, $\delta(0) = 1$, $\frac{d\delta}{dx} \geq 0$ (in our model, $b(x) = 1 - \frac{x}{K}$, $\delta(x) = 1$). The results of this study remain qualitatively the same as long as some density dependence of the division rate is present (so, $db/dt$ is not identically zero). In these cases, we have $\mathscr{F}(\tau) < 1$ if $\tau > 1$ and $\mathscr{F}(\tau) > 1$ if $\tau < 1$, leading to the mutants becoming strongly negatively (positively) selected under higher (lower) turnover. In the case where $b(x)=1$ (and the death rate grows with population size) it appears that the coarse-grained approach developed here must be modified to incorporate deme extinction. Understanding further aspects of the generalized system is the subject of ongoing work.

## Discussion

In this paper, we have demonstrated that population structure fundamentally alters the evolutionary dynamics of mutants, which are characterized by proportionally scaled birth and death rates. In well-mixed populations, such quasi-neutral mutants exhibit fixation probabilities that deviate from neutrality but remain distinct from truly advantageous or disadvantageous mutants. However, in spatially

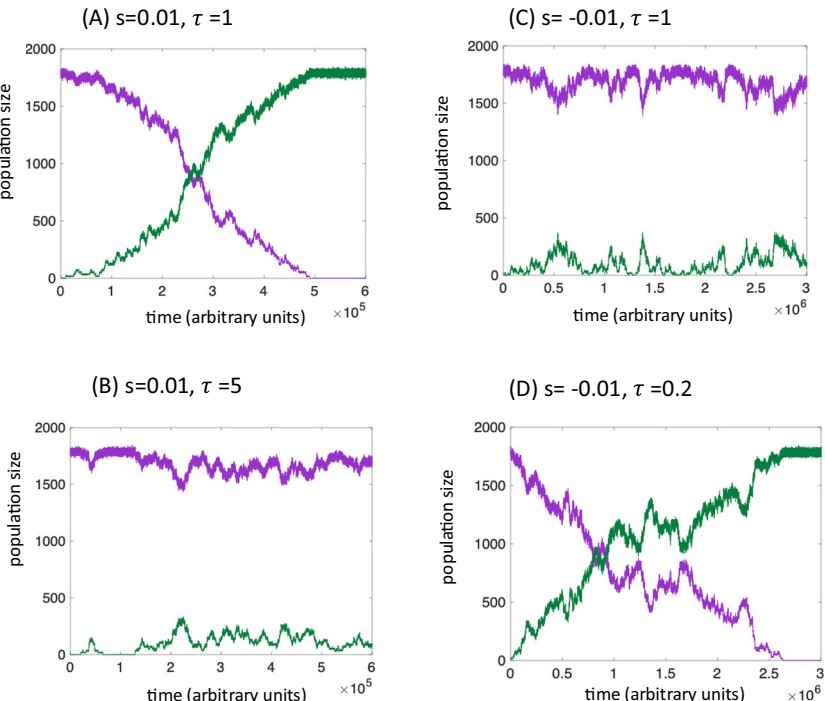

**Fig. 3 | Proportionally scaled birth and death rates can reverse selection in the deme-structured model with mutations.** Time-series of the wild-type (purple) and mutant (green) numbers are shown for typical simulation realizations. **A** A mutant with a 1% advantage ($s = 0.01$) is assumed, without any scaled birth and death rates compared to the wild-type ($\tau = 1$). As expected, the mutant invades. **B** Same 1% advantage, but the mutant simultaneously has 5-fold higher turnover compared to the wild-type ($\tau = 5$), and this renders the mutant disadvantageous, persisting at a selection-mutation balance. **C** A mutant with a 1% disadvantage ($s = -0.01$) is assumed, without any scaled birth and death rates compared to the wild-type ($\tau = 1$). As expected, the mutant persists at selection-mutation balance. **D** Same 1% disadvantage, but the mutant simultaneously has 5-fold reduced turnover compared to the wild-type ($\tau = 0.2$), leading to a selective advantage. Baseline parameter values were as follows: $r_w = r_m = 5$; $d_w = d_m = 0.5$; $\epsilon = 10^{-4}$; $K = 20$; $\mu = 10^{-6}$.

structured or fragmented populations, these dynamics shift dramatically: mutants with increased turnover (higher birth and death rates) experience negative selection, while those with decreased turnover (lower birth and death rates) gain a selective advantage. This effect leads to the reversal of selection, whereby mutants with a higher maximal life-time reproductive output become negatively selected if their divisions and deaths are accelerated, a strong effect that only increases with the population size. Similarly, mutants with a smaller maximal life-time reproductive output become positively selected if their divisions and deaths are decelerated. These effects are driven by the interplay between demographic fluctuations and spread in structured populations, highlighting the critical role of spatial organization in evolutionary outcomes.

We obtained an analytical approximation for the probability of mutant fixation in a fragmented population, where

P(fix in a fragm. pop.)=P(fix of a mutant in a deme) x P(fix of a mutant deme).

For the probability of fixation of a mutant deme in a population of demes, we have

$$P(\text{fix of a mutant deme}) = \frac{1 - \frac{1}{\mathscr{F}}}{1 - \frac{1}{\mathscr{F}^D}}.$$

This is the result of a coarse-grained approximation, and formally, the quantity $\mathscr{F}$ plays the role of a relative mutant deme fitness (please note that this is formally the fixation probability in a Moran process with relative mutant fitness $\mathscr{F}$). This quantity is very useful, especially given that it has a remarkably simple form, e.g., $\mathscr{F} = \tau \frac{\bar{\rho}_m}{\bar{\rho}_w}$ for quasi-neutral mutants, where $\bar{\rho}_m$ and $\bar{\rho}_w$ are, respectively, the expected probability of mutant fixation in a wild-type deme and the expected probability of wild-type fixation in a mutant deme, starting from a single individual (similar expressions are also derived for other scenarios).

Importantly, this result is not related to the concept of group selection, for the following reason: the fact that the quantity $\mathscr{F}$ enters the probability of fixation in this way is not an assumption of the model at the level of demes, but rather a mathematical consequence of the individual-level selection in the presence of rare migrations. Even though individual reproductions are driving the dynamics, if the migration rate of individuals from one deme to another is relatively low (as assumed by the coarse-grained model) then most demes will contain just one of the types: either they are populated by wild-type individuals, or by mutant individuals. Only very rarely does one find demes that contain both wild-type and mutant individuals at the same time. In such a scenario, the question becomes: how many mutant demes arise from a given deme that contains mutant individuals? This is where the "fitness of a deme" comes in, which accounts for the surprising mutant fixation probabilities observed in our spatial computer simulations.

This framework can be contrasted with the models of multilevel selection considered in a number of papers e.g., refs. 19,20, where, on top of rules of individual dynamics, additional rules governing deme divisions were assumed. These rules relate the deme behavior as a whole to the state of the individuals in the deme. In our case, no such assumptions are made, and what we see is simply a mathematical simplification that takes a very intuitive form.

It is further instructive to compare our results with the findings of Bhat & Guttal[13], who investigated the phenomenon of selection reversal in well-mixed populations (see also Supplementary Note 1, Section 1.4). Both in fragmented populations studied here and in non-fragmented populations[13], effects connected with fitness (such as changes in lifetime reproductive potential) and effects resulting from changes in turnover trade-off shape the selective pressure experienced by a mutant. The major qualitative difference is that in well-mixed systems, the effect of turnover disappears with higher population

sizes, and in fragmented populations the influence of turnover becomes more pronounced for larger populations. For example, if the selection coefficient of the mutant is positive but its kinetic rates are accelerated ($s > 0$, $\tau > 1$), in a large well-mixed system this mutant will behave as advantageous, and its fixation probability will approach a constant as $N_{tot}$ increases. On the contrary, in a fragmented system, it will have characteristics of a disadvantageous mutant with P(fix) decaying exponentially with the system size.

The finding that the life-time reproductive success of individuals does not necessarily correlate with Darwinian fitness in structured populations influences interpretation of evolutionary processes across many biological systems. On the biomedical side, tumor progression can involve the emergence of cells characterized both by an increased proliferation and death rate[21–27] in spatially structured cell populations. Even if the overall reproductive success of these mutant cells is higher than that of the original cells, their chances to emerge can be reduced and the time to invasion increased compared to established evolutionary theory. Similar considerations apply to faster replication rates of viruses, which can be associated with faster viral decay rates or higher death rates of infected cells[28–30]. Conversely, a selective disadvantage (e.g., of a drug-resistant mutant before therapy[31]) might be substantially reduced (or even overturned) if the fitness cost of a resistant mutant is associated with a reduced overall turnover, which can be important when predicting standing genetic variation in a population.

In ecological systems, it is well-established that there can be tradeoffs between reproduction and life-span of individuals. Such reproductive tradeoffs are well-documented in female arthropods, and evidence has further accumulated that this also applies to males[10]. Among higher animals, pregnancy can result in reduced mobility and escape velocity during predator attacks in a variety of organisms[3–8]. This can lead to an increased death rate in pregnant individuals, which has been demonstrated experimentally e.g., for the mosquitofish *Gambusia affinis*[9]. In these cases, the conditions for the selection of new variants can be more complicated than indicated by existing evolutionary theory, and not related to reproductive success in a straightforward way. Our results can be further relevant for invasive species, for example, the rapid population growth of the invasive snail species *Melanoides tuberculata* in new habitats can be countered by a high mortality rate due to crayfish-induced predation[5]. This theory helps us better understand how this competition plays out, and how this impacts management strategies.

## Methods

### The deme-structured model

Stochastic simulations were performed to study deme structured dynamics, see also ref. [32]. Consider $D$ demes, with the numbers of wild-type and mutant individuals in each deme, $i$, are given by $\{x_i, y_i\}$, where $x_i$ and $y_i$ are non-negative integers. The reproduction and death rates are denoted by $r_w$ and $d_w$ for wild-type individuals and $r_m$ and $d_m$ for mutants, and the carrying capacity is denoted by $K$. We assume $r_m = (1 + s) \tau r_w$, $d_m = \tau d_w$, where $\tau$ represents the scaling of mutant birth and death rates, and $s$ the selection coefficient ($s = 0$ for quasi-neutral mutants). The simulation proceeds as follows. First, the total event propensity is calculated for each deme: $A_i = (r_w x_i + r_m y_i)\left[1 - \frac{x_i + y_i}{K}\right]_+ + (d_w x_i + d_m y_i)$, where $[...]_+$ denotes the positive part. Denoting the total propensity $A = \sum_{i=1}^{D} A_i$, we start each update by selecting a deme, where the probability to pick deme $i$ is given by $A_i/A$. Suppose deme $k$ is chosen. Then the event in that deme is selected according to the following probabilities: wild type division with probability $r_w x_k \left[1 - \frac{x_k + y_k}{K}\right]_+ / A_k$; wild type death with probability $d_w x_k / A_k$; mutant division with probability $r_m y_k \left[1 - \frac{x_k + y_k}{K}\right]_+ / A_k$; mutant death with probability $d_m y_k / A_k$. If a wild type or mutant death event is chosen, then the value of $x_k$ or $y_k$ is decreased by one. If a wild type division is chosen, then with probability $1-\epsilon$ the value of $x_k$ is increased by one, and with probability $\epsilon$ another deme is picked at random, uniformly. If that target deme (say deme $j$) has

the total population below the carrying capacity ($x_j + y_j < K$), then $x_j$ is increased by one representing the migration of a (new-born) wild type individual from deme $k$ to deme $j$; otherwise, the event is aborted (representing the density controlled migration model). The division and migration of mutant individuals is handled in the same way. Finally, the time-step duration that corresponds to the update is chosen as an exponentially distributed random variate with the mean $1/A$.

We note that this Gillespie type simulation corresponds to the basic deterministic dynamics in each deme: $\dot{x}_i = r_w x_i (1 - \frac{x_i + y_i}{K}) - d_w x_i + \text{migr}_{x,i}$, $\dot{y}_i = r_m y_i (1 - \frac{x_i + y_i}{K}) - d_m y_i + \text{migr}_{y,i}$, where terms $\text{migr}_{x,i}$ and $\text{migr}_{y,i}$ stand for the appropriate migration processes. We further note that several different models of migration have been implemented (see Section 1.3.2 of Supplementary Note 1); the algorithm described above is migration model (2a), crowd-controlled migration by division: after a cell division, the daughter cell migrates with probability $\epsilon$ to a randomly chosen deme, but the event is aborted if the target deme is at or above its carrying capacity. Other models are implemented in a similar fashion.

To study the probability of mutant fixation, the Gillespie simulation was initiated with $K$ wild-type cells in all $D$ demes, and run for 10,000 time-steps to reach a quasi-equilibrium. Then a single mutant individual was added to a randomly chosen deme if the population size in that deme was below carrying capacity. Otherwise, a new deme was chosen until the mutant was placed successfully. The simulation was then continued until either the mutant or wild-type population went extinct, and the outcome was recorded.

To study systems with de-novo mutations, we included the process of mutation by assuming, with probability $\mu$, that one of the offspring cells in every wild type division is mutant. A sample computer code is given in Supplementary Code 1.

### Spatial, stochastic agent-based model

We consider a grid that contains $n \times n$ spots, which may be either empty or contain a wild-type or mutant individual. At each time step, the grid is sampled randomly until $M$ spots that contain individuals have been found (where $M$ denotes the current number of nonempty spots). If a chosen spot is nonempty, there is a probability for the individual to die (with probabilities $D_w$ and $D_m = \tau D_w$ for wild-types and mutants, respectively), and there is a probability to attempt a reproduction event with probabilities $R_w$ and $R_m = (1 + s) \tau R_w$, respectively. If an individual is chosen for reproduction, one of the 8 nearest neighboring spots is selected randomly, and if that spot is empty, the offspring individual is placed there (and reproduction does not proceed otherwise). Periodic boundary conditions were assumed. To start the simulation, the average wild-type population size when fluctuating around steady state was determined numerically, and this number of individuals was randomly seeded across the grid. The simulation was run for 500 time-steps to allow the spatial densities to equilibrate, and then a randomly chosen wild-type individual was replaced with a mutant. The simulation was run until either the mutant or wild-type population went extinct, and the outcome was recorded. To study systems with de-novo mutations, the process of mutation with probability $\mu$ was included for wild type division events. A sample computer code is given in Supplementary Code 1.

### Reporting summary

Further information on research design is available in the Nature Portfolio Reporting Summary linked to this article.

## Data availability

This study was based on mathematical and computational modeling and does not contain data.

## Code availability

Computer codes that were used to obtain the results are available as supplementary materials, see Supplementary Code 1.

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

## Acknowledgements

We would like to acknowledge funding from National Science Foundation grant DMS 2424853 (D.W. and N.K.), and DMS 2435484 (D.W. and N.K.).

## Author contributions

N.K. and D.W. conceptualized the research, performed the analysis, and wrote the paper.

## Competing interests

The authors declare no competing interests.
