## [Transparent Peer Review file · Nature Communications]

Population structure reverses selection of variants with proportionally scaled birth and death rates

Corresponding Author: Professor Dominik Wodarz

Version 0:

Reviewer comments:

Reviewer #1

(Remarks to the Author)

This paper reports the interesting observation that mutants which have larger turnover, but are traditionally considered to be neutral, can be under selection from the change in turnover rate alone. This observation makes sense in the context that the authors put it into, but traditional thinking would not lead to this conclusion. I consider this an important insight that is certainly worth to be published.

However, I do have a couple of comments:

Major:

- From reading the main text, it was not possible for me to fully grasp what kind of model the authors have in mind. I understand that they may get carried away from their excitement of the finding and felt forced to write this “for a general audience”, but I think it crucial for this kind of paper to clearly explain the model at the start. In the methods section, the authors describe an ODE and that they perform “stochastic Gillespie-type simulations” based on this – but to me, many stochastic processes can correspond to a single ODE. It would be more meaningful to me to mention the elementary processes and their rates and I would need that knowledge before trying to understand the models. So far, this crucial information is only part of the supplement. Please consider adding a more detailed description of the model to the main text – I think a general audience should be able to digest this.
- In line 89, the concept of “fitness of a mutant deme” is introduced – I do not think this is necessary here, as all processes are triggered by individual reproduction. Otherwise, this is dangerously close to the concept of “group fitness”, which can be problematic.
- Line 117: fixation probability is simply equal to s – to me, this is not that clear. For the Moran model, it is $s/2$, for the Fisher-Wright model it is s – why is it clear that it is s here?
- In the SI, I was wondering if the model would change if the carrying capacity induced reduction of the birth rate would instead be a carrying capacity induced increase of the death rate. Would the conclusions remain valid? I do not think that it is necessary to develop this idea in detail, but it would be good to discuss it to figure out how robust the conclusions of the authors are.

Minor:

- Abstract and discussion „more frequent pregnancy can be accompanied by a higher chance to die due to predation in ecological systems” – the switch from viruses and cells to pregnancy and sexual reproduction is quite abrupt to me. Anything in between?
- In Fig. 1, the scales are not ideal – e.g-. 10^{-5} , 10^{-4} ,... would make it clear that we deal with a logarithmic scale, in the inset of Fig. 1A this is hardly readable.
- Line 96, “such that a diffusion approximation is valid” – this is unclear. When is an approximation “valid”? And does diffusion approximation not usually imply that selection reduces with increasing N ?
- Line 110: please define the range of s , it seems that $-1 \leq s \leq \infty$.
- The authors use “simply” quite extensively. This has great disruptive potential for the uninitiated reader.
- Line 151: Is it clear that tumor progression always leads to an increased proliferation and death rate? Or could it just affect one of these parameters?
- SI; after Eq. 12: Is the order 12,10,11 deliberately chosen here?

(Remarks on code availability)

It is great that the authors immediately share their code (in Fortran 90! I think it is a good idea to publish this original code, probably the next generation of scientists will use a LLM to translate to Julia or Python – but this should not be the concern of the code developer if others prefer other languages). However, I did not go through the code or tried to run it.

Reviewer #2

(Remarks to the Author)

The paper presents a counterintuitive theoretical prediction that in metapopulations, an advantageous mutation can turn into a disadvantageous mutation (and vice versa) as the turnover rate increases (decreases), even when the mutant and the wild type have identical fitness. The authors argue that this effect becomes even more pronounced in spatially structured populations. The concepts developed in this paper are a spatial extension of the recent AmNat paper by Bhat & Guttal (2025).

Overall, I think the paper is well written and makes a novel theoretical argument, which can have implications for cancer biology and invasion dynamics.

My main comment about the manuscript relates to how fitness and turnover rate are defined. In a logistic type model, the Malthusian fitness of a 'type' should be $w_i = (r_i - d_i)$, and the turnover rate should be $(r_i + d_i)$ (Bhatt and Guttal 2025). However, (I think) the authors define fitness as the ratio of r_i and d_i and regulate the turnover rate by τ , such that $w_m = \tau * (r_w / d_w)$.

This becomes important because the authors are trying to separate the effect of differential fitness from differential turnover in determining the fixation probability. One of their principal arguments is that a mutant with lower turnover might be favoured even when the wild type and mutant have identical fitness. If the fitness and turnover rate are defined as $(r_i - d_i)$ and $(r_i + d_i)$, then $w_m = \tau * (r_w - d_w)$, and the turnover rate is $\tau * (r_w + d_w)$. As a result, changing τ changes both fitness and turnover rate.

Some clarification would be nice. I should acknowledge that I have not worked with master equations before, so I could not fully evaluate the mathematics in the supplementary information.

The paper's title is slightly misleading. The noise-induced selection concept also applies to a single deme case (Fig. 1A). The authors show that spatial structure magnifies this effect even more. Therefore, suggesting that population structure reverses selection is not correct.

"Population structure magnifies the reversal of selection due to demographic stochasticity" or something similar might be a better title.

Minor comments

I understand Nature Communications is a short-format journal, but a discussion of previous work on noise-induced selection might be a good idea. The authors do cite some papers (10-12). However, as it stands, it appears that the authors are the first to report this phenomenon.

Line 64: spatially structured magnifies the noise-induced selection

(Remarks on code availability)

Version 1:

Reviewer comments:

Reviewer #1

(Remarks to the Author)

The authors have addressed all my comments carefully and I am generally happy with the revision and recommend publication.

However, I would like to comment on two of my previous suggestions. I leave it upon the authors / the editor if any changes are implemented, though:

1. The description in the model is to me hidden in the section "materials and methods" (which is an utterly strange name imposed on the theoretical community by experimentalists – maybe you can just name it "model" here?). I personally would start with the description of the basic processes (currently line 354) and only later mention the ODEs. After all, the focus is on the Gillespie simulations and the ODEs just capture them in a particular limiting case. But this may be a matter of taste.

2. I think I have to explain the (my?) confusion about the approximation of the Moran formula (admittedly, I had to plot the three different approximations and explore a bit when the s is valid and when $1/N+s/2$ is a better approximation): The authors perform $N \rightarrow \infty$ first and then they look at small s , approximating the fixation probability by s . This is a meaningful approach, but I feel the comparison with the original Moran framework is always a bit awkward, as N is entirely absent in the equation and one can hence not explore the interplay between s and N . It seems that this holds for $sN \gg 1$. Another possibility is to think of small s from the start and do Taylor series of the entire, exact fixation probability – leading to something like $1/N + s/2 + \dots$ - and now considering large N . This implies dropping the $1/N$ term (keeping in mind that $sN \ll 1$), which would lead to $s/2$. Thus, I feel there is some justification for the confusion. Another minimal remark: At least in my version of the book of Warren Ewens, chapter 5.3 is on diffusion approximation, so the approximation is a bit more cumbersome than just dropping the term with N and doing a Taylor approximation. Probably there is a more direct calculation for the $1/s$ result for the Moran process in the book, but I did not find it so far.

Please apologize that I am so picky here!

(Remarks on code availability)

Reviewer #2

(Remarks to the Author)

The authors have adequately addressed my point of concern. I have no further comments.

(Remarks on code availability)

Reply to the reviewers

We thank the reviewers for reading the paper and for providing thoughtful comments. The comments have been very useful for us and enabled us to significantly revise and improve the manuscript. All reviewer comments were taken into account in the revision. Please find our point-by-point reply here. For convenience, we append a manuscript and supplement version with tracked changes below the point-by-point reply.

Reviewer #1 (Remarks to the Author):

This paper reports the interesting observation that mutants which have larger turnover, but are traditionally considered to be neutral, can be under selection from the change in turnover rate alone. This observation makes sense in the context that the authors put it into, but traditional thinking would not lead to this conclusion. I consider this an important insight that is certainly worth to be published.

However, I do have a couple of comments:

Major:

- From reading the main text, it was not possible for me to fully grasp what kind of model the authors have in mind. I understand that they may got carried away from their excitement of the finding and felt forced to write this “for a general audience”, but I think it crucial for this kind of paper to clearly explain the model at the start. In the methods section, the authors describe an ODE and that they perform “stochastic Gillespie-type simulations” based on this – but to me, many stochastic processes can correspond to a single ODE. It would be more meaningful to me to mention the elementary processes and their rates and I would need that knowledge before trying to understand the models. So far, this crucial information is only part of the supplement. Please consider adding a more detailed description of the model to the main text – I think a general audience should be able to digest this.

We appreciate this comment and agree. We have now explained the stochastic process considered here (“Gillespie simulations”) in detail in Materials and Methods of the main text, including the description of the algorithm and the simulation procedure.

- In line 89, the concept of “fitness of a mutant deme” is introduced – I do not think this is necessary here, as all processes are triggered by individual reproduction. Otherwise, this is dangerously close to the concept of “group fitness”, which can be problematic.

The reviewer is raising a very important issue, which we have been pondering over. We think that the fitness of a mutant deme is insightful here, because it’s a quantity (\mathcal{F}) that enters the formula

$P(\text{fix in a fragm. Pop.})=P(\text{fix in a deme}) \times P(\text{fix of a mutant deme}),$

where

$P(\text{fix in a fragm. Pop.}) = \rho_m$ and

$P(\text{fix of a mutant deme}) = \frac{1 - \frac{1}{\mathcal{F}}}{1 - \frac{1}{\mathcal{F}D}}$,

where the latter is the probability of fixation of a mutant deme in a population of demes. This is basically the main result of our coarse-grained approximation, and formally, the quantity \mathcal{F} plays the role of a relative mutant deme fitness (please note that this is simply the fixation probability in a Moran process with mutant fitness \mathcal{F}). So in this sense, this quantity is enlightening, especially given that it has such a simple form: $\mathcal{F} = \tau \frac{\rho_m}{\rho_w}$.

Importantly, this is not group selection, for the following reason: the fact that the quantity \mathcal{F} enters the probability of fixation in this way is not an assumption of the model at the level of demes, but rather a mathematical consequence of the individual-level selection in the presence of rare migrations. Even though individual reproductions are driving the dynamics, if the migration rate of individuals from one deme to another is relatively low (as assumed in our study) then most demes will contain just one of the types: either they are populated by wild-type individuals, or by mutant individuals. Only very rarely do you find demes that contain both wild-type and mutant individuals at the same time. In such a scenario, the question becomes: how many new wild-type demes arise from a given deme that contains wild-type individuals? How many mutant demes arise from a given deme that contains mutant individuals? This is where the “fitness of a deme” comes in, which accounts for the surprising mutant fixation probabilities observed in our spatial computer simulations.

This can for example be contrasted with the models of multilevel selection considered in e.g. [Traulsen, A., & Nowak, M. A. (2006). Evolution of cooperation by multilevel selection. *Proceedings of the National Academy of Sciences*, 103(29), 10952-10955.] and [Henriques, G. J., van Vliet, S., & Doebeli, M. (2021). Multilevel selection favors fragmentation modes that maintain cooperative interactions in multispecies communities. *PLOS Computational Biology*, 17(9), e1008896.], where, on top of rules of individual dynamics, additional rules about deme divisions were assumed. These rules relate the deme behavior as a whole to the state of the individuals in the deme. In our case, no such assumptions are made, and what we see is simply a mathematical simplification that takes a very intuitive form. We have now explained this in more detail, and set it apart from group selection, to avoid any possible confusion, see the Discussion section of the revised paper.

- Line 117: fixation probability is simply equal to s – to me, this is not that clear. For the Moran model, it is $s/2$, for the Fisher-Wright model it is s – why is it clear that it is s here?

We were referring to the usual Moran formula, whereby the probability of mutant fixation

(starting from 1 mutant) is given by $\frac{1 - \frac{1}{1+s}}{1 - \frac{1}{(1+s)^N}} \approx \frac{s}{1+s} \approx s$, where the first approximation

corresponds to $s \gg 1/N$ and the second to $s \ll 1$. This result is still a good approximation in the presence of demographic fluctuations. We have now clarified this in the text, providing two references.

- In the SI, I was wondering if the model would change if the carrying capacity induced reduction of the birth rate would instead be a carrying capacity induced increase of the death rate. Would the conclusions remain valid? I do not think that it is necessary to develop this idea in detail, but it would be good to discuss it to figure out how robust the conclusions of the authors are.

We thank the reviewer for this insightful question. We have chosen the density-dependent division model because it more closely matches the spatially explicit simulations; it also is arguably a more biologically realistic assumption as cells slow down their divisions in crowded conditions. One way to generalize the results of the fragmented model is to assume that the per capita division rate is $rb(x)$ with $b(0)=1$, $\frac{db}{dx} \leq 0$, and the per capita death rate is $d\delta(x)$, $\delta(0) = 1$, $\frac{d\delta}{dx} \geq 0$ (in our paper, $b(x) = 1 - \frac{x}{K}$, $\delta(x) = 1$). The results of this study remain qualitatively the same as long as some density dependence of the division rate is present (so, db/dx is not identically zero). In these cases we have $\mathcal{F}(\tau) < 1$ if $\tau > 1$ and $\mathcal{F}(\tau) > 1$ if $\tau < 1$, leading to the mutants becoming strongly negatively (positively) selected under higher (lower) turnover. In the case where $b(x)=1$ (and the death rate grows with population size) it appears that the coarse-grained approach developed here must be modified to incorporate deme extinction. One complication is that the demes size distribution is a lot wider (say, with $b(x) = 1$, $\delta(x) = 1 + \frac{x}{K}$) and individual populations are more likely to go extinct as the deme population size of the fragmented system decreases. In the new version of the manuscript we added a discussion of these points. Understanding further aspects of the generalized system is subject of ongoing work.

Minor:

- Abstract and discussion „more frequent pregnancy can be accompanied by a higher chance to die due to predation in ecological systems” – the switch from viruses and cells to pregnancy and sexual reproduction is quite abrupt to me. Anything in between?

Yes, that is true. We now have provided intermediate examples and revised the text in the abstract, introduction, and discussion.

- In Fig. 1, the scales are not ideal – e.g. 10^{-5} , 10^{-4} ,... would make it clear that we deal with a logarithmic scale, in the inset of Fig. 1A this is hardly readable.

We have changed the axes marking in the inset, to make it more easily recognized as a log scale, and modified the inset to make it more visible.

- Line 96, “such that a diffusion approximation is valid” – this is unclear. When is an approximation “valid”? And does diffusion approximation not usually imply that selection reduces with increasing N ?

We thank the reviewer for bringing up these points. While selection usually is reduced as the total population size increases, in our case N refers to a deme size in a model where the total population size, N_{tot} , remains constant. In addition, in our case, the selection coefficient is zero ($s=0$, quasi-neutral mutants) and selection-like pressure comes about through an unusual mechanism, which requires fragmentation into demes that are sufficiently small.

The text explaining the connection with the diffusion approximation was unclear in the previous version. In the revised version, we pointed out that expression $p_{fix} = \frac{1}{N} \times \frac{2}{\tau+1}$ was derived under the diffusion approximation. Further, we replaced the text around line 96 as follows:

“Suppose the total population of size N_{tot} is split into D demes of size N each. Decreasing individual deme size N (and therefore increasing D) will magnify the effect, thus making accelerated (decelerated) mutants less (more) advantageous. If, on the other hand, the demes are large, the effect disappears: for large N , the deviation from the diffusion approximation is negligible, and we have $\frac{\rho_m}{\rho_w} = \frac{1}{\tau}$, thus resulting in $\mathcal{F}=1$ (deme neutrality) and $\Pi_1^{frag} = \frac{1}{DN} \times \frac{2}{\tau+1}$, which is exactly the same as the probability of fixation in a non-fragmented system (of size $N_{tot}=DN$).”

- Line 110: please define the range of s , it seems that $-1 \leq s \leq \infty$.

We have now defined the range of s , and also clarified what biologically realistic values s might attain.

- The authors use “simply” quite extensively. This has great disruptive potential for the uninitiated reader.

We agree and have re-formulated those sentences (including the Supplement).

- Line 151: Is it clear that tumor progression always leads to an increased proliferation and death rate? Or could it just affect one of these parameters?

We agree, this was not formulated correctly. There are examples where tumor cells are characterized both by an increased proliferation and death rate, as shown in the referenced papers. It is not a universal feature. Instead of saying “tumor progression involves the emergence of cells...”, we now wrote “tumor progression can involve the emergence of cells... as shown experimentally...”

- SI; after Eq. 12: Is the order 12,10,11 deliberately chosen here?

Yes, this order is chosen because equation (12) replaces equations (8,9) and corresponds to $i=1$, while equation (10) is for $1 < i < K$, and equation (11) is for $i=K$. We have explained this in the new text.

Reviewer #2 (Remarks to the Author):

The paper presents a counterintuitive theoretical prediction that in metapopulations, an advantageous mutation can turn into a disadvantageous mutation (and vice versa) as the turnover rate increases (decreases), even when the mutant and the wild type have identical fitness. The authors argue that this effect becomes even more pronounced in spatially structured populations. The concepts developed in this paper are a spatial extension of the recent AmNat paper by Bhat & Guttal (2025).

Overall, I think the paper is well written and makes a novel theoretical argument, which can have implications for cancer biology and invasion dynamics.

My main comment about the manuscript relates to how fitness and turnover rate are defined. In a logistic type model, the Malthusian fitness of a 'type' should be $w_i = (r_i - d_i)$, and the turnover rate should be $(r_i + d_i)$ (Bhatt and Guttal 2025). However, (I think) the authors define fitness as the ratio of r_i and d_i and regulate the turnover rate by τ , such that $w_m = \tau * (r_w / d_w)$.

This becomes important because the authors are trying to separate the effect of differential fitness from differential turnover in determining the fixation probability. One of their principal arguments is that a mutant with lower turnover might be favoured even when the wild type and mutant have identical fitness. If the fitness and turnover rate are defined as $(r_i - d_i)$ and $(r_i + d_i)$, then $w_m = \tau * (r_w - d_w)$, and the turnover rate is $\tau * (r_w + d_w)$. As a result, changing τ changes both fitness and turnover rate.

Some clarification would be nice. I should acknowledge that I have not worked with master

equations before, so I could not fully evaluate the mathematics in the supplementary information.

We thank the reviewer for raising this interesting and important issue. We agree that the concept of “fitness” is not straightforward, and there are many functional definitions in the literature. One must choose the one that serves the purpose of answering the biological question at hand.

It does not appear that defining fitness, for type i , as $r_i - d_i$, would make sense in our context (here, r_i and d_i are per capita constant rates of divisions and death, evaluated in the absence of density dependence). Indeed, in the deterministic limit, the ODE for type i would be

$$\dot{x}_i = r_i x_i \left(1 - \frac{\sum x_k}{K}\right) - d_i x_i$$

where $\sum x_k$ denotes the total population. Assume only 2 types (this analysis generalizes to an arbitrary number of types); the equilibria are:

$$x_1 = K(1 - d_1/r_1), x_2 = 0 \text{ and } x_1 = 0, x_2 = K(1 - d_2/r_2).$$

Unless $r_1/d_1 = r_2/d_2$, there is a globally stable equilibrium corresponding to competitive exclusion of the species with the smaller r/d . In this case neutrality is determined by the ratio r/d , which is the maximum lifetime reproductive output of a type (and is similar to basic reproductive number, R_0 , in epidemiology).

On the other hand, species that have the same value of $r-d$ do not have to be neutral. For example, take two species with the same value of $r-d$:

$$r_1 = 10, d_1 = 9.1, \quad r_1 - d_1 = 0.9, \quad \frac{r_1}{d_1} \approx 1.1 \quad (\text{Example 1})$$

and

$$r_2 = 1, d_2 = 0.1, \quad r_2 - d_2 = 0.9, \quad \frac{r_2}{d_2} = 10$$

The second species will take over in this model, because it has a larger r/d ratio. (This is also the case for spatial generalizations of this density-dependent model).

There is, however, no contradiction with the paper by Bhat & Guttal (2025). In their formalism, Malthusian fitness of type i is defined as $b_i(x) - d_i(x)$, where the two terms are density-dependent per capita division and death rates. For our model this would correspond to defining fitness as

$$w_i = r_i \left(1 - \frac{\sum x_k}{K}\right) - d_i$$

This quantity correctly describes the result of competition if one starts from an equilibrium corresponding to the dominance of one of the types (say, type 1). Evaluated at $x_1 = K(1 - d_1/r_1), x_2 = 0$, we have

$$w_1 = 0, \quad w_2 = r_2 \frac{d_1}{r_1} - d_2 = \frac{d_1 d_2}{r_1} \left(\frac{r_2}{d_2} - \frac{r_1}{d_1}\right) \quad (**)$$

so again, the type with a larger r/d will have a higher fitness. To illustrate this, consider the following example:

$$r_1 = 10, d_1 = 9, \quad r_1 - d_1 = 1, \quad \frac{r_1}{d_1} \approx 1.1 \quad (\text{Example 2})$$

and

$$r_2 = 1, d_2 = 0.1, \quad r_2 - d_2 = 0.9, \quad \frac{r_2}{d_2} = 10$$

According to (**), we have $w_1 = 0$ and $w_2 > 0$, so species 2, which has a higher r/d ratio, will win the competition, which is correct.

Applying this definition of Malthusian fitness at low numbers, however, is not as useful. We have $w_1 = r_1 - d_1, w_2 = r_2 - d_2$, which ascribes a larger fitness to the first type in numerical Example 2 ($r_1 = 10, d_1 = 9$). This type is characterized by a faster initial growth, but will eventually lose the competition. So Malthusian fitness, being simply the per capita (possibly population-dependent) growth rate, is not in the general case a predictor of competition outcomes.

In our paper, instead of talking about Malthusian fitness and turnover, we mostly rely on the maximum lifetime reproductive output (r_i/d_i) and turnover factor (e.g. $\tau = d_2/d_1$). Assume that

$$r_2 = \tau r_1, d_2 = \tau d_1$$

(the mutant's maximum lifetime reproductive output is the same as that of the wild type). For instance, take $\tau = 0.1$, and set

$$r_1 = 10, d_1 = 1, \quad r_1 - d_1 = 9, \quad \frac{r_1}{d_1} = 10 \quad (\text{Example 3})$$

and

$$r_2 = 1, d_2 = 0.1, \quad r_2 - d_2 = 0.9, \quad \frac{r_2}{d_2} = 10$$

Calculating the Malthusian fitness at the wild-type equilibrium for Example 3, we have (see (**)) $w_1 = w_2 = 0$, as expected. Therefore, scaling the kinetic rates by a turnover factor does not affect the Malthusian fitness at the equilibrium. This is also consistent with previous works by (Parsons et al 2010). Please note that Malthusian fitness (the growth rate) at low numbers is affected by this scaling and is higher for the faster turning mutant. This was discussed in our earlier paper (Wodarz et al, Evolutionary Applications 2017): faster turning mutants grow faster initially, but then lose competition once density-dependence kicks in.

We are thankful to the referee for the opportunity to discuss these points. A comparison of our model with that of by Bhat & Guttal (2025) is included in the revised manuscript, please see the new section 1.4 of the Supplement.

The paper's title is slightly misleading. The noise-induced selection concept also applies to a single deme case (Fig. 1A). The authors show that spatial structure magnifies this effect even more. Therefore, suggesting that population structure reverses selection is not correct.

“Population structure magnifies the reversal of selection due to demographic stochasticity” or something similar might be a better title.

We thank the reviewer for bringing up these points. We agree that selection can be reversed even in a single deme, which we did not emphasize in the previous version of the paper. This phenomenon is comprehensively described in (Bhatt and Guttal 2025) and was previously noted in [Kaveh, K., Komarova, N. L., & Kohandel, M. (2015). The duality of spatial death–birth and birth–death processes and limitations of the isothermal theorem. *Royal Society open science*, 2(4), 140465.] in the context of the (single-deme) Moran process.

In the revised text, we have now included a discussion of our results in light of those of (Bhatt and Guttal 2025), please see the new Figure 2(C,D) and the surrounding text in the Results section, and also the Discussion section, where we compare and contrast this effect in a single vs fragmented/spatial populations. In particular, in a single deme the effect disappears as N increases, which also follows from the scaling of the turnover term derived in (Bhatt and Guttal 2025), from the diffusion approximation in (Parsons et al 2010), and from explicit formula (4.13) in [Kaveh et al 2015]. Remarkably, in a fragmented population the effect of suppression increases with population size. In fact, in a wide range of parameters, selection suppression is not observed in a single deme, but is very strong in a fragmented population of the same size. We illustrate this with additional figure 2(C,D).

We would not like to include “demographic stochasticity” in the title because selection reversal happens in the death-birth Moran process in the absence of demographic stochasticity (constant population). We have therefore modified the title to “Population structure reverses selection of variants with proportionally scaled birth/death rates, an effect increasing with the population size”.

Minor comments

I understand *Nature Communications* is a short-format journal, but a discussion of previous work on noise-induced selection might be a good idea. The authors do cite some papers (10-12). However, as it stands, it appears that the authors are the first to report this phenomenon.

We absolutely agree with the reviewer. In the revised version of the manuscript, we have included a much extended discussion of previous work, where we show how our contribution is building upon the literature, and exactly how it advances the field. In particular, please see Introduction, where we write:

“In the recent years it has become apparent that the interplay between birth and death rates can influence the evolutionary dynamics in subtle ways, particularly when these kinetic rates are proportionally scaled across competing variants. A striking example of this phenomenon is the

behavior of quasi-neutral mutants—variants whose birth and death rates are scaled by a common factor, preserving their maximum lifetime reproductive output but altering their turnover rates. Parsons et al.¹⁰ studied a birth-death process with demographic fluctuations, and demonstrated that in well-mixed populations, quasi-neutral mutants exhibit fixation probabilities that deviate from strict neutrality. In particular, faster-turnover mutants are less likely to fixate due to increased demographic stochasticity, and slow turnover mutants are more likely to fixate. This finding challenged classical neutral theory by showing that even in the absence of fitness differences, proportional scaling of kinetic rates alone could shape evolutionary outcomes.

Similar patterns were observed in other processes. For example, the role of cell cycle acceleration and deceleration in evolutionary dynamics was studied¹³. Two life strategies were compared: repair-efficient cells that enter temporary cell cycle arrest (corresponding to repair) and repair-deficient cells that do not arrest (and thus cycle faster). It was found that although non-arresting cells were characterized by a faster growth in isolation, this did not translate into a selective advantage in the model: faster cycling cells fixated with a probability that was lower than predicted for a neutral scenario. It was further shown that similar patterns are observed in a death-birth Moran process, where faster (slower) mutants fixated at a lower (higher) rate compared to neutral.

A recent paper by Bhat & Guttal¹² offers a comprehensive study of the interplay between Malthusian fitness and turnover in well-mixed populations. Their analysis revealed how noise-induced selection can reverse the direction of expected evolutionary trajectories. The authors derive stochastic differential equations from a general nonlinear birth-death process to describe changes in population densities and trait frequencies, revealing a drift term containing a balance between natural selection for increased ecological growth rate and a stochastic selection for reduced variance in changes in population densities, which favors species with a lower turnover. The latter mechanism can reverse the direction of selection predicted by deterministic models, particularly in small populations or under weak selection. This approach highlighted the dual mechanisms by which demographic stochasticity biases evolutionary outcomes: directly, by favoring the variants with a higher Malthusian fitness, and indirectly, through noise-related effects that depend on the turnover rate. The work further generalized classical equations like the replicator-mutator equation, Price equation, and Fisher's fundamental theorem to finite populations, providing a framework to understand how stochasticity interacts with natural selection and drift.

In this study, we build on these theoretical advances by exploring how population structure, such as deme or spatial organization, further modulates the evolutionary fate of mutants with proportionally scaled kinetic rates. While previous work focused on well-mixed populations, we demonstrate that structured environments fundamentally alter selection pressures. Variants with proportionally increased birth and death rates, which are quasi-neutral in well-mixed settings, become truly disadvantageous in structured populations, and vice versa for decreased rates. Moreover, we show that even mutants with a higher lifetime reproductive output can be rendered disadvantageous by accelerated turnover, an effect that in spatially structured populations grows with the population size.”

We have also included a discussion of our results in light of (Bhatt and Guttal 2025) in the Results section (figure 2 and the surrounding text), the Discussion section, and Supplementary Materials, Section 1.4.

Line 64: spatially structured magnifies the noise-induced selection

We could not find this sentence in the manuscript...

Population structure reverses selection of variants with proportionally scaled birth/death rates, an effect increasing with the population size

We thank the reviewers for the comments and have integrated them into the manuscript. This file contains our point-by-point reply, and we also append manuscript and supplement versions with changes tracked below the reply.

Reply to the referee report

Referee #1

The authors have addressed all my comments carefully and I am generally happy with the revision and recommend publication.

We thank the referee for this positive assessment.

However, I would like to comment on two of my previous suggestions. I leave it upon the authors / the editor if any changes are implemented, though:

1. The description in the model is to me hidden in the section “materials and methods” (which is an utterly strange name imposed on the theoretical community by experimentalists – maybe you can just name it “model” here?). I personally would start with the description of the basic processes (currently line 354) and only later mention the ODEs. After all, the focus is on the Gillespie simulations and the ODEs just capture them in a particular limiting case. But this may be a matter of taste.

We agree with the referee that it makes more sense to start the model description from the stochastic process and then mention the ODEs. We have now rearranged the Methods section (the section title changed from “Materials and Methods in accordance with instructions from the Journal), and the model description now follows this logic.

2. I think I have to explain the (my?) confusion about the approximation of the Moran formula (admittedly, I had to plot the three different approximations and explore a bit when the s is valid and when $1/N+s/2$ is a better approximation): The authors perform $N \rightarrow \infty$ first and then they look at small s , approximating the fixation probability by s . This is a meaningful approach, but I feel the comparison with the original Moran framework is always a bit awkward, as N is entirely absent in the equation and one can hence not explore the interplay between s and N . It seems that this holds for $sN \gg 1$. Another possibility is to think of small s from the start and do Taylor series of the entire, exact fixation probability – leading to something like $1/N + s/2 + \dots$ - and now considering large N . This implies dropping the $1/N$ term (keeping in mind that $sN \ll 1$), which would lead to $s/2$. Thus, I feel there is some justification for the confusion. Another minimal remark: At least in my version of the book of Warren Ewens, chapter 5.3 is on diffusion approximation, so the approximation is a bit more cumbersome than just dropping the term with N and doing a Taylor approximation. Probably there is a more direct calculation for the $1/s$ result for the Moran process in the book, but I did not find it so far.

We completely agree with the referee that these points require a more careful treatment. We have now created a new subsection in the Supplement (called “Review of mutant fixation probability in the Moran process”, section 1.1.2), where we present the Moran probability of fixation formula (with references) and derive the large- \$N\$ limit under the assumptions that \$N|s| \gg 1\$ and \$|s| \ll 1\$, for both disadvantageous and advantageous mutants. We also provide the formula derived by Kimura by using the diffusion approximation, where both limits can be easily derived: if \$s < 0\$, \$N \rightarrow \infty\$ then the probability of fixation decays exponentially with \$N\$, and if \$s > 0\$, \$N \rightarrow \infty\$, the probability of fixation approaches a constant. The difference is a factor of 2 that appears in Kimura and Haldane papers, because they used the Fisher-Wright formulation of the process; this is a known difference in the two formalisms that stems from the overlapping vs non-overlapping generations in the two processes. We agree that Ewen’s Chapter 3 did not

provide this result explicitly, so we replaced that reference with the references to Kimura and Haldane, and also explicitly derived the results for the Moran process.

Referee #2

The authors have adequately addressed my point of concern. I have no further comments.

We thank the referee for their guidance with this paper.